# Culturally Adaptive Governance—Building a New Framework for Equity in Aboriginal and Torres Strait Islander Health Research: Theoretical Basis, Ethics, Attributes and Evaluation

**DOI:** 10.3390/ijerph18157943

**Published:** 2021-07-27

**Authors:** Daniel L.M. Duke, Megan Prictor, Elif Ekinci, Mariam Hachem, Luke J. Burchill

**Affiliations:** 1Faculty of Medicine, Dentistry and Health Sciences, University of Melbourne, Melbourne, VIC 3010, Australia; Daniel.duke@unimelb.edu.au (D.L.M.D.); elif.ekinci@unimelb.edu.au (E.E.); mariam.hachem@unimelb.edu.au (M.H.); 2Melbourne Academic Centre for Health (MACH), Melbourne, VIC 3010, Australia; 3Health, Law and Emerging Technologies Programme, Melbourne Law School, University of Melbourne, Melbourne, VIC 3010, Australia; megan.prictor@unimelb.edu.au; 4Centre for Digital Transformation of Health, Faculty of Medicine, Dentistry and Health Sciences, University of Melbourne, Melbourne, VIC 3010, Australia

**Keywords:** Indigenous governance, ethics, adaptive governance, critical allyship

## Abstract

Indigenous health inequities persist in Australia due to a system of privilege and racism that has political, economic and social determinants, rather than simply genetic or behavioural causes. Research involving Aboriginal and Torres Strait Islander (‘Indigenous’) communities is routinely funded to understand and address these health inequities, yet current ethical and institutional conventions for Indigenous health research often fall short of community expectations. Typically, mainstream research projects are undertaken using traditional “top-down” approaches to governance that hold inherent tensions with other dominant governance styles and forms. This approach perpetuates long-held power imbalances between those leading the research and those being researched. As an alternative, Indigenous governance focuses on the importance of place, people, relationships and process for addressing power imbalances and achieving equitable outcomes. However, empowering principles of Indigenous governance in mainstream environments is a major challenge for research projects and teams working within organisations that are regulated by Western standards and conventions. This paper outlines the theoretical basis for a new Culturally Adaptive Governance Framework (CAGF) for empowering principles of Indigenous governance as a prerequisite for ethical conduct and practice in Indigenous health research. We suggest new orientations for mainstream research project governance, predicated on translating theoretical and practical attributes of real-world ethics, adaptive governance and critical allyship frameworks to Indigenous health research. The CAGF is being implemented in a national Indigenous multicenter trial evaluating the use of continuous blood glucose monitors as a new technology with the potential to improve diabetes care and treatment for Indigenous Australians—the FlashGM Study. The CAGF is a governance framework that identifies the realities of power, acknowledges the complexities of culture and emerging health technologies, and foregrounds the principle of equity for mainstream Indigenous health research.

## 1. Introduction

Aboriginal and Torres Strait Islander (respectfully hereafter ‘Indigenous’ (Unless distinguishing both Aboriginal and Torres Strait Islander cultures and Peoples is deemed important to the point at hand)) health research operates in an environment of ambiguity and is beset with ethical and practical deficiencies. While ethical guidelines have been developed for Indigenous health research over the past 35 years, little attention has been paid to the organisational and governance environments in which these guidelines seek articulation, rendering key kinds of power invisible and perpetuating long-held inequities in Australia. We suggest new orientations for mainstream research projects centred around a Culturally Adaptive Governance Framework (CAGF) for equity in Indigenous health research (The terms ‘mainstream’ and ‘Western’ will be used in this paper; however, we do acknowledge their limitations and the numerous and often vague applications of the terms do not capture the distinctive characteristics of a range of cultural traditions that often tends to condense Indigenous/Western perspectives and culture. We subscribe to arguments around its problematic use made by Linda Tuhiwai Smith in ‘Decolonizing methodologies: Research and Indigenous peoples’ (2012, P. 93), who draws on arguments made by Stuart Hall articulating the idea of the ‘West’ as a concept and language for imagining a set of stories, ideas, historical events, social relationships that functions in ways which (1) allow ‘us’ to characterise and classify societies into categories, (2) condense complex images of other societies through a system of representation, (3) provide a standard model of comparison, and (4) provide criteria of evaluation against which other societies can be ranked. However—where used in this paper for convenience—the term simply helps facilitate the framing of obvious differences in cultural and intellectual traditions between Anglo and Indigenous Australians). The proposed CAGF offers an integrated framework that encompasses ethics, governance, consent and allyship as a missing link between ethical guidelines, review processes and equitable research outcomes. It is an approach that fulfils organisational and project level commitments to successful, real-world ethics, by acknowledging the realities of power, the complexities of culture, and the rise of emerging health technologies, while highlighting the principle of equity as a necessary foundation for ethical conduct and practice in Indigenous health research.

This paper begins by foregrounding the historical struggle for reform that first generated ethical guidelines for health research involving Indigenous communities and continues to draw them into question, arguing that we are still far from genuinely engaging with community concerns and priorities in health research. On this basis, we present a new framework of research governance—the CAGF. It then defines governance, a term often not well understood or adequately defined. In doing so it elucidates three governance modes—hierarchical, market and network—that compete in research governance, as well as defining Indigenous governance as a distinct form of governance. We then define and pair adaptive governance with critical allyship in forming the theoretical basis for the CAGF and as essential for enabling conditions suitable for empowering principles of Indigenous governance in research. We ground this with a real-world consequentialist ethics overlay that is posited as an additional layer to ethical guidelines and review processes. The paper then goes on to outline three defining attributes considered necessary for successful implementation of the framework. A systemic evaluation framework is then detailed as an integral element for implementing and maintaining structural elements of the CAGF in research. Finally, we highlight the reform agenda that generated the guidelines as an unfinished project, one that is yet to see the broader transformation of Indigenous health research through Indigenous control as an imperative for self-determination. The CAGF is intended as a meaningful, replicable tool for research teams to empower principles of Indigenous governance and thus a vehicle for increasing Indigenous voices within and control over health research in Australia.

## 2. Materials and Methods

### 2.1. Ethical Guidelines

In 1986, the national conference on ‘Research Priorities in Aboriginal Health’ was convened in Alice Springs jointly by the National Health and Medical Research Council (NHMRC) and the Menzies Foundation. Since then, the primacy of ethical conduct in health research with Indigenous Peoples in Australia has become far more salient. Final recommendations from the 1986 conference included the need for greater community control, more culturally appropriate research methods, and improved flow of knowledge and benefits arising from research [1].

These recommendations became the reference point for early iterations of NHMRC guidelines [2], but are still far from being fully realised [3], reflecting a failure to decisively capture the mood of transformational reform of the late 1980s. A year after the Alice Springs conference, Indigenous representatives at a Camden workshop put forward ‘Principles, Standards and Rules’ arguing that the creation of ‘guidelines’ were no guarantee of bringing about broader attitudinal change in non-Indigenous researchers or improved research practices [4]. The NHMRC, nonetheless, promulgated ‘ethical guidelines’ in 1991, with these and subsequent iterations retaining significantly Western notions of researcher independence within research projects and a mainstream institutional dominance over the funding and conduct of health and medical research more broadly [5]. The absence of formal mechanisms for Indigenous oversight of research projects once they fall within the parameters of mainstream conventions also speaks to a lack of attention given to conceptualising what comes ‘after the guidelines’ [6].

Consequently, research remains a site of struggle for Australia’s First Peoples and ethical and research conventions for Indigenous health research continue to fall short of community expectations [7]. The articulation of ethical guidelines within mainstream research conventions continues to be an amalgam of Western standards regulating Indigenous knowledge and values. This has resulted in a research industry that encourages procedural observance of ethical guidelines rather than critical evaluation and adaption of research practices to better reflect the values and standards of Indigenous communities [6]. The central importance of research governance for successful translation of ethical guidelines to the conduct and practice of research also appears to be missing. The word ‘governance’ is absent from the latest version of the NHMRC ethical guidelines (2018) [8] and the historical struggle for self-determination, so fundamental to the development of the guidelines has been relegated to a link on the NHMRC website, which at the time of writing is broken.

Presently, there does not appear to be any framework that can genuinely empower Indigenous leadership and governance within already-funded research projects. Furthermore, longstanding issues surrounding the way Western intellectual approaches conceptualise Indigenous knowledge and culture as static and anchored in the past continue to limit the emancipatory potential of dynamic, complex, and ever-changing Indigenous knowledges and cultures across time and place. Considering this, it is crucial to attend to the lack of appropriate governance frameworks for achieving the ideals set out in the various guideline documents. To address this gap, we propose the CAGF, which elevates the final recommendations of the 1986 conference as central to achieving ethical conduct in Indigenous health research.

While it is still too early to assess the impact of the Australian Institute of Aboriginal and Torres Strait Islander Studies (AIATSIS) new ‘Code of Ethics’, released late 2020, far more emphasis on suitable governance frameworks as an imperative for more ethical research is clearly evident in this updated code [9]. The word ‘governance’ appears 27 times, and on page 7 it states “All research must be conducted with ethics and integrity. Institutions that are regularly undertaking or sponsoring research have a responsibility to develop research governance frameworks to support this”. We consider this to be highly instructive and posit that this shift in focus towards research governance frameworks that support ethical conduct and practice in Indigenous health research has paved the way for the CAGF as an integrated framework that can be used by mainstream organisations to advance health equity in this setting. We turn now to consider the concept of governance and different governance styles and forms relevant to Indigenous health research.

### 2.2. Governance

The concept of governance has continually been hard to define. Context, system logic and values all create vastly different defining characteristics (Table 1). However, broadly speaking, “the concept of governance aims at capturing the complexity of real-world policy processes” surrounding the rules and actions within a system and how they are regulated, structured and held accountable [10] (p. 85). Within a governance system there are different types of actors or modes, who have different roles, which are defined in terms of their system logic and the nature of interactions that they undertake. We have adopted the classical distinction recognising three major governance modes: hierarchies, markets, and networks as outlined by Pahl-Wostl and others (Table 2) [11,12,13]. For other key terms refer to Table 1.

All three modes are identifiable and exert distinct influence within the system logic of Indigenous health research projects. Researchers belong to research organisations that invariably operate within hierarchical structures. Funding for projects is competitive and successful grants come with market-style forces that are transactional and time sensitive, threatening to undermine partnerships with community stakeholders that rely heavily on investments of time to develop genuine relationships built around trust. This complicates the regulation and structure of project governance initiatives as forces often compete, adding complexity to the overall logic, values, and ethics of each project. The responsibility to manage these tensions and the contrasting expectations from funding agencies, institutional conventions and guideline documents falls largely to research teams and researchers who enter Indigenous spaces—often without the necessary training, experience and supports. Current ethical guidelines, rather than bridging these divides, add further complexity of network style forces that individual research projects must navigate.

### 2.3. Indigenous Governance

In defining governance as a concept, it is also important to distinguish Indigenous governance as a distinct form of governance; one necessarily dependent upon some normative aspects of the concept of governance but nevertheless predicated upon a distinct knowledge and value system that sits outside traditional notions of governance. Indigenous governance therefore adds further complexity to the overall system logic in health research in Australia.

For tens of thousands of years, Indigenous peoples in Australia have used their own processes and structures to govern, adapt and organise themselves in a system of geographies to express their culture, values and traditions [14]. While these processes and structures remain vital elements of Indigenous governance in the present day, Indigenous peoples’ understanding of governance is informed by their experience of colonisation and the continuing struggle for self-determination in Australia. Yawuru leader and barrister, Professor Mick Dodson states “when we speak of Indigenous governance we are not referring to the pre-colonial state. Rather, we are referring to contemporary Indigenous governance: the more recent melding of our traditional governance with the requirement to effectively respond to the wider governance environment” [15] (p. 89). Indigenous governance then, reflects contemporary Indigenous culture, values and traditions expressed as a desire for contemporary self-determination and a need to interface this with broader governance structures of the Australian state.

Indigenous governance is therefore both a political expression of the need to reclaim authority to determine how and by whom narratives that reflect Indigenous culture, values and realities are expressed, and institutional arrangements and technicalities. It reflects a need to interface practically and effectively with organisational governance structures, but also stipulates that the principle of good governance requires Indigenous communities to have genuine decision-making powers in matters pertaining to their lives and realities. The Indigenous Community Governance Project (ICGP), which was carried out between 2004 and 2008, defined governance in this context as “the evolving processes, relationships, institutions and structures by which a group of people, community or society organise themselves collectively to achieve the things that matter to them”. It also stressed that Indigenous governance was “as much about people, power, and relationships as it is about formal structures, management and corporate technicalities” [16] (p. 9).

With traditional governance arrangements for Indigenous health research failing to adequately accommodate competing forces and styles of governance, Indigenous governance successfully accommodates both formal and informal structures and styles of governance and does so in a way that is relevant to Indigenous people today. It is therefore vital that Indigenous governance be empowered and prioritised in Indigenous health research; however, we acknowledge that without an appropriate framework in which to give space and meaning to principles of Indigenous governance in mainstream arenas, arrangements are likely to fall back to more traditional and dominant mainstream governance conventions for research. We therefore posit that an adaptive approach to governance, one that puts Indigenous governance, knowledge and culture at its heart, is the most appropriate model in this context. The following section will articulate two key theoretical underpinnings of the CAGF we are proposing: adaptive governance and critical allyship.

### 2.4. Theoretical Frameworks

#### 2.4.1. Adaptive Governance

Adaptive governance has emerged in recent years as an analytical and practical framework for the holistic management of complex problems and governance environments [17]. Underpinning its emergence, especially within the field of socio-ecological and climate systems, has been a need for robust governance arrangements that respond flexibly and adaptively in complex and dynamic environments. As an emerging framework of governance, it addresses the repeated failures of traditional governance and management systems to respond effectively in highly uncertain contexts [18].

The approach is defined by a view that governance and management in complex systems needs to be able to respond with resilience in the face of irreducible uncertainty, all the while building sustainable community and governance capacities that are attentive to broader social, cultural, and technological contexts in planning, implementation, and evaluative processes. This is seen as especially relevant for governance environments that incorporate different stakeholders from different levels and modes, such as private, state and community actors. Meaningful collaboration is required across these actors and scales, which relies heavily on self-organising capabilities and partnerships, characterised by both formal and informal means of establishing and sustaining meaningful networks and relationships. These essential governance processes are viewed as crucial for effective decision-making, power sharing and the production of new knowledge in an adaptive governance approach [19].

#### 2.4.2. Critical Allyship

Critical allyship is a practice that guides the “actions of people in positions of privilege for resisting the unjust structures that produce health inequities” [20] (p. 1). Health inequities are defined as health disparities that are systematically upheld between groups with different levels of underlying social advantage and disadvantage [21]. As an intersectional framework for recognising privilege in structures and positionings, it is considered a solidarity approach for those who want to reorient their positions from ‘saving’ to critical allyship with those who are most disadvantaged by these structures. Importantly, critical allyship is not an identity, but an ongoing practice or orientation appropriate for recognising one’s privilege and working with an understanding that inequalities and disadvantage are produced within a system of various structural constraints. In this sense, critical allyship is a personal process of unlearning and learning that is consistent, active and demanding, where those in positions of privilege seek to re-evaluate how to work in solidarity with those from disadvantaged groups of people.

Indigenous health inequities persist in Australia due to a system of privilege and racism that has political, economic, and social determinants, rather than simply genetic or behavioural causes. Much Indigenous health research and health care continues to be framed in ways that do not take into consideration these broader structural constraints, limiting the transformative potential of any health inquiry or action [22]. Using a critical allyship approach requires critical evaluation and reorienting from this dominant way of thinking about health inequities and research. This can be achieved through focusing on the impact of one’s actions, rather than on one’s intent, and requires an understanding of one’s position of privilege within a system that sustains those in relative positions of oppression and is upheld by dominant ideologies and assumptions that pervade government, legal, health care, education, and financial systems, as well as long-held attitudes and norms [23].

Research connecting mainstream institutions with Indigenous communities is a site of engagement in the processes of two-way learning and knowledge exchange, which has been referred to as the ‘cultural interface’, between mainstream and Indigenous knowledge systems [24]. This cross-cultural space is complex and dynamic and requires research teams and organisations to reflexively examine their own beliefs, judgments, and practices and how these may differ from those being researched. This reflexive examination must extend beyond merely their conduct in a defined project by also being attentive to the broader social and cultural contexts in which their research is invested. For non-Indigenous researchers seeking to engage in this space effectively and positively it entails challenging assumptions about the nature of their understanding of reality (ontology) and the ways in which they construct that reality, especially regarding the methods used (epistemology), as well as challenging the values and ethical judgements that underpin this reality (axiology).

The concept of ally or allyship has been critiqued by Indigenous scholars for its over- and misuse, and its commodification as a social identity that serves to maintain colonial hierarchical structures [25,26]. In this context, ally relationships with those who are disadvantaged tend to be romanticised and tied to social identity, but which reassert paternal relationships that negates the title or intent of an ally. As Andrea Sullivan-Clarke of the Muskogee Nation of Oklahoma states, “Relationships are a fundamental part of Indigenous philosophies… Thus, an Indigenous epistemology often includes a normative component—the universe is moral, and all of our relations provide knowledge as to how to live… Indigenized conceptions of allyship should be understood as a relationship that promotes the well-being of those being served”. In this sense, the critical allyship we advocate is allyship that also considers Indigenous critiques of allyship for developing, investing in, and sustaining relationships with those they seek to serve. Importantly, this critical allyship respects decision-making processes of Indigenous communities and supports Indigenous Peoples affirmation of their sovereignty within colonial structures by understanding the long history of struggle that often-included serious violence and which contributes to ongoing mistrust. It also establishes allyship, and the relationships invested in, as critical processes of rebuilding community trust and goodwill.

#### 2.4.3. Culturally Adaptive Governance for Indigenous Health Research

Culturally Adaptive Governance is a framework for research governance bringing together principles of adaptive governance and critical allyship to advance the principle of equity in Indigenous health research by genuinely investing in Aboriginal and Torres Strait Islander knowledge systems, values, governance and leadership. It is defined as a system of research governance that is flexible, adaptive, integrative, and responsive to the complexities of Indigenous cultures in Australia. It is a framework for facilitating shifts in power and to the prevailing research system logic by acknowledging the incongruence of Western standards regulating Indigenous values and norms. It suggests polycentric structures of governance that empower community concerns, needs and priorities in Indigenous health research.

The CAGF is intended to support Indigenous and non-Indigenous stakeholders engaged in Indigenous health research through a critical allyship approach that facilitates cross-cultural dialogues, knowledge and resource exchange. This will enable research teams to be more attentive and responsive to the changing concerns, needs and priorities of the Indigenous communities they engage in research. It provides those Indigenous people who choose to engage in research with a continuing say in how research is conducted in their communities across the life of a project, recognising that much research with Indigenous people to date has not resulted in tangible benefits for the communities involved [7]. It encourages stakeholders to identify opportunities to build capacity, capability, opportunities, and outcomes, for Indigenous research partners throughout the life of the research project. In this sense, Indigenous governance principles are supported and centred by the CAGF, rather than running in tension or rendered peripheral to the overall system logic of research governance. This differs from previous attempts to highlight and address deficiencies in ethical research with Indigenous communities, as the CAGF outlines a holistic framework to support the alternative research methodologies outlined in ethical guidelines and contained in Indigenous research reform agendas.

#### 2.4.4. Adaptive Governance as a Vehicle for Indigenous Governance

Translating theoretical and practical attributes of adaptive governance to Indigenous health research for what we believe is the first time, the development of the CAGF is a response to repeated failures of traditional governance structures to bring about sustainable ethical, equitable and beneficial research outcomes for Indigenous Peoples in Australia. A major reason for these failures has been the dominance of Western intellectual and moral epistemological and ontological approaches to health and medical research at the expense of Indigenous governance and knowledge systems [27].

Adaptive governance models have been put forward as viable alternatives and responses to the failures of traditional governance and management regimes in achieving sustainable success and enhanced community capacity through instances of complex change and uncertainty [28]. They also stipulate the need to empower alternative ways of thinking about a complex problem in order to better address it [17,19,29]. Yet, Indigenous ways of knowing, being and doing continue to be situated as marginal or peripheral to mainstream conventions, rather than fundamentally enmeshed in core ways of undertaking research. Fundamental to epistemological inequality in Indigenous health research is the failure to recognise Indigenous knowledge and culture in Australia as complex, dynamic and everchanging across time and space.

According to Flavier and colleagues, “Indigenous Knowledge is the information base for a society, which facilitates communication and decision-making. Indigenous information systems are dynamic and are continually influenced by internal creativity and experimentation as well as by contact with external systems” [30] (p. 479). Aboriginal and Torres Strait Islander knowledge systems are complex and reflect the diverse histories, spatial, geographic, and cultural identities of Australia’s First Peoples. In this sense, Indigenous knowledge, rather than being easily codified as a theory, is more experiential, being reinforced through trial, error and continuing lived experiences [31].

### 2.5. Ethical Framework for the CAGF

Recognising that ‘governance’ itself is not a panacea for deficiencies in Indigenous health research, the CAGF uses ethics as an ‘overlay’ for research governance and as a way of facilitating more ethical and equitable arrangements and decisions between stakeholders engaged in Indigenous health research. This overlay of real-world ethics for research governance is in addition to guidelines governing ethical conduct and practice at the national level (i.e., NHMRC Guidelines, AIATSIS Code of Ethics), as well as regional or community frameworks such as the South Australian Research Accord [32] and the Inala Community Jury model [33]. Taking this approach, the CAGF strengthens the ability of research teams to engage in ethical conduct and practices that reflect shared values of a collaborative network. It also prepares teams for ethical decision making in response to unexpected events including major changes in the broader socio-political-ecological context in which research is being conducted, a recent example being the decision by many research organisations to pause or adjust research activities in response to the COVID-19 pandemic.

The CAGF guides stakeholders to consider what ethical decision-making is and why it is essential within research governance and organisational environments over and above the ethical framework provided by guideline documents. In this sense, it rejects a narrow view of ethics represented by the guidelines, preferring instead to empower consequentialist models of ethics that give weight and attention to the consequences of any decision or action in determining right or wrong [34]. Using such a model enables three questions to be established relating to consequences, benefits and judgement.

What are the consequences of any action/decision?Who is/are the primary beneficiaries of this action/decision?How and by whom ought the consequences of any decision/action to be judged?

Susan Liautaud outlines a further three pillars that we believe support this ethical approach to decision making [35]:Transparency—the open sharing of important information.Informed consent—agreeing to an action based on an understanding of the action and its consequences.Effective listening—grasping the meaning.

Many of these principles overlap with the theoretical underpinnings of adaptive governance and critical allyship and as such we believe that rather than being processes of ethics (as with ethical guidelines and review boards), successful ethical practice overlays every action and decision undertaken in research.

### 2.6. Defining Attributes of the CAGF

A systematic literature review carried out in 2018 by Sharma-Wallace and colleagues identified eight defining methods required for successfully implementing adaptative governance initiatives [36]. We consider all eight to be indicative of a desired adaptive governance system logic. However, for the purposes of focusing on the most vital elements of ethical Indigenous health research, we adapt their eight methods into three defining attributes we consider to be representative of a desired system logic of the CAGF, which we call Invest, Effect and Foster.

Invest in community priorities and Indigenous leadership capacity;Effect purposeful collaboration between Indigenous knowledge systems and Western scientific traditions;Foster goodwill and meaningful connections.

#### 2.6.1. Invest—Investing in Community Priorities and Indigenous Leadership Capacity

Investing time and resources into identifying and realising community priorities within the scope of research projects will enable principles of Indigenous governance and self-determination to be at the heart of research governance arrangements. It will also lead to greater community control of research, the development of more culturally appropriate research methods, and improved flow of knowledge and benefits arising from research—as put forward by Indigenous representatives at the 1987 Camden workshop. Careful consideration should be given to the scope of management required within research governance structures and what resources will be necessary for appropriate and scalable community action to be achieved, by connecting governance functions to specific community strengths and contexts. This is especially important, as the stakeholders most in need for an adaptive approach have often been the least able to benefit from them due to differences in resources [36].

Investing in community priorities and engagement also involves recognising and using community-level frameworks and resources, incorporating them into formal project management plans, acknowledging community and broader societal histories and events as integral to community realities and inclusion of appropriate community members into meaningful roles within implementation teams and management boards. Indigenous researchers, community partners and stakeholders should be meaningfully involved in all decision-making forums and processes across the life of a research project, and this also requires research teams to recognise that what might work for one community may not necessarily translate to working for another. The capacity to promote effective Indigenous leadership at all levels of research projects and across a variety of stakeholders is also essential for reframing entrenched research governance protocols and arrangements, while an overdependence on a small number of Indigenous leaders will also have adverse effects on governance and project arrangements, and on long-term sustainable outcomes.

#### 2.6.2. Effect—Effecting Purposeful Collaboration between Indigenous Knowledge Systems and Western Scientific Traditions

Indigenous communities are principally defined by connection to country and understanding this connection along with the diversity of Australia’s First Peoples is critical for anyone undertaking Indigenous health research. This also requires an understanding of the history of colonisation and Indigenous health research in Australia and how these histories inform ongoing tensions arising from Western standards regulating Indigenous values and norms. Effecting purposeful collaboration between differing knowledge systems and traditions can only be supported using forums or ‘bridges’ that connect problems with potential solutions regardless of the different levels at which actors or stakeholders are positioned. To operate in complex and dynamic environments, stakeholders require access to the best available information. Therefore, informal and formal feedback loops are essential for guiding and understanding decision-making processes and for identifying tensions arising from competing knowledge systems within research governance systems.

In this sense, the CAGF should be viewed as a robust framework of equity and inclusion needed within processes of project design and implementation that elucidates the nuanced and contextual definitions of ‘community’; definitions that do not succumb to homogenous conceptualisations of Aboriginal and Torres Strait Islander people [37]. Nyunggai leader Warren Mundine states “Indigenous Australians are united as kindred spirits in their shared history. But we will always be separate mobs first and foremost with our distinct and unique country, heritage, languages and cultures” [38]. This diversity is often not adequately acknowledged or defined within research conventions that too often rely solely on individual Indigenous leaders and/or advisory groups for community oversight and voice. The Indigenous Governance Toolkit defines ‘community’ as “a network of people and organisations linked together by webs of personal relationships, cultural identities, political connections, traditions and rules, shared histories, social and economic conditions and/or common understandings and interests”. It further acknowledges three different types of communities: discrete geographic communities, dispersed communities of identity and a community of interest.

The CAGF recognises both the diversity of Indigenous communities and the existence of different types or contextual layers of community that are encompassed within research environments. It stipulates a need for step-by-step processes of engagement through the different layers of community, which are purposeful in instigating cycles of learning and bridge building and focused on empowering Indigenous governance and self-determination. This helps establish Indigenous communities as dynamic, complex and everchanging across the life of a research project and reminds research teams that Indigenous communities aren’t fixed or static. Relationships and effective lines of communication are essential to respond to changes at the community level, including those that may be outside the primary context of research.

Using the CAGF, building relationships begins with First Nations people working across the research stakeholder network. These may include Indigenous organisational leaders, Indigenous researchers, and/or community level health workers or project officers participating in research program delivery. Connections should then be made with community organisations and peak advocacy groups, ensuring they are aware of research being conducted and given an opportunity to comment or collaborate. Community members are also engaged and given a ‘seat at the table’ to share lived experiences of the health issue that is the research focus. The combination of these diverse voices generates an Indigenous voice for the research project that helps ensure research is conducted ethically and in accordance with community expectations, including the identification of tangible benefits to community (Figure 1). Engagement is necessarily ongoing across the different layers of community and potentially extends beyond the life of the research project. Feedback from First Nations people on how the research is meeting the needs of community members is used to keep research on track with the potential for local adaptations in response to local needs. This signifies a deliberate shift away from purely advisory or consultative roles for Indigenous people to roles with genuine decision-making capacity and authority within Indigenous health research.

It is also vital to allow appropriate timeframes for establishing trust via ongoing dialogues between Indigenous communities and research teams. Continuing dialogues are necessary to facilitate knowledge exchange between Indigenous communities and researchers leading to a common understanding of what resources (knowledge, personnel, infrastructure, funding) and changes in governance structures are needed to address community needs in ways that are meaningful to community and researchers. Indigenous communities need to be empowered to question the research and point out when it does not make sense or appear to be of value. Agency must be given to Indigenous communities to propose alternative hypotheses and questions that better reflect Indigenous world views, methodologies, and priorities. Where equity in this context is not properly considered or defined, this has been found to limit collaborative decision-making, defaulting governance arrangements to dominant top-down political governance structures and styles [36]. Centralised, top-down governance arrangements that fail to engage with local communities or understand local contexts and priorities and empower Indigenous researchers will likely fail to enable co-design of effective and sustainable governance and research solutions.

Indeed, Indigenous researchers and leaders are integral to the CAGF, with their particular networks and methodologies given support by the adaptive nature of the CAGF. This is intended to protect Indigenous representation from assuming the load of maintaining and centring Indigenous voice and governance principles throughout the life of the project. This is achieved through a focus on critical allyship with non-Indigenous researchers. Enabling relationships that prioritise Indigenous knowledge and decision making across all forums within the research network, with community voices continually sought using cycles of feedback and learning that ensures community perspectives reflect those that represent the community. In this sense, Indigenous researchers and leaders are not expected to speak to or provide oversight to all things ‘Indigenous’ and much more emphasis is placed on seeking out strong relationships and community voices of authority, thus allowing Indigenous researchers to assume much more appropriate roles within research collaborations that focus on their professional, cultural and methodological expertise.

#### 2.6.3. Foster—Fostering Goodwill and Meaningful Connections

Underpinning meaningful connections should be base levels of trust, goodwill and familiarity between actors that will form the basis for building social capital within governance and project outcomes. Bound up in identifying local priorities and investing time for significant local engagement, meaningful connections are developed incrementally and over extended periods of time and achieved through both formal and informal modes of collaboration [39]. Addressing uneven power dynamics should also be positioned as central for ensuring equal participation and contribution between stakeholders in collaborative forums. So-called ‘institutional memory’ should also be prioritised for new stakeholders entering the collaboration, with effective governance and collaboration outcomes expedited through physical resources or financial goodwill gestures from research teams to communities involved [36].

Key to forming meaningful connections in Indigenous health research is an approach that is relational and not transactional. Indigenous and non-Indigenous stakeholders require time to relate according to country, community and culture. Values are shared and the motivation for undertaking Indigenous health research should be discussed. These connections are supported by a critical allyship framework that reorients from the position of ‘saving’ to undertaking critical allyship with those who are most disadvantaged by the structures that those with privilege commonly work within, and benefit from. It is for this reason that ongoing evaluation is vital for identifying and exploiting opportunities for achieving the desired system logic of the CAGF. Outcomes or results of adaptive governance initiatives have been relatively difficult to evaluate, measure, maintain and replicate, especially in the face of ongoing uncertainty and change [40]. Therefore, robust and comprehensive planning, preparation and monitoring is essential to ensure alignment between stakeholder values, support articulation of ethical principles and practice, and the broader project aims through continual cycles of feedback and analysis. It is an evaluation framework for formal and informal feedback and analysis cycles that the paper now outlines.

### 2.7. Evaluating the Implementation and Impact of the CAGF

As we have outlined, the CAGF provides a framework for shifting research project governance from the dominant Western ways of conducting research towards a new desired system logic. Such a shift requires the CAGF itself to undergo continual cycles of evaluation and analysis across the life of the research project. This evaluation needs to be comprehensive and capture the complexity of multiple stakeholders working within the CAGF. Central to these processes are feedback mechanisms that enable continuing cycles of analysis and learning across and between stakeholders engaged in Indigenous health research. This learning informs effective decision making and promotes flexibility and integrated responses to community concerns, needs and priorities over time.

To capture the inherent complexity of the CAGF, a systemic research design drawing on the work of Pahl-Wostl and Lebel will be used to evaluate its effect on implementation and research outcomes in Indigenous health research projects. This systemic approach to research design is considered across four categories: Structure, Dynamics, Context, Indicator based (Table 3) [41]. Particular attention is given to the contextual conditions within each category or case study, which are especially relevant given the cross-cultural and critical allyship contexts in which the CAGF seeks articulation.

The governance structure (social networks, organisations, other systems) is mapped using diagrams that highlight power, influence, authority, responsibilities, and accountabilities across the research collaboration. Dynamics within a research collaboration focuses on the dynamics of change within a research governance structure. Dynamic methods pay particular attention to the various forms of competing governance forces at play, enabling the identification of deficiencies to the overall system logic for equitable and beneficial community centred outcomes. They also attempt to highlight opportunities for instigating shifts towards the desired system logic and articulating indicators for measuring success or failure in achieving the desired state.

Contextual approaches are necessarily qualitative in nature and use a combination of context-specific surveys, qualitative interviews, and case studies. These questions seek to understand stakeholders’ understanding of the CAGF, their perception of its ability to achieve its intended aim of promoting more equitable Indigenous health research, and any challenges experienced working within an Indigenous health research project using the CAGF. Outcomes arising from the use of the CAGF are also captured with a particular emphasis on flow of benefits to Indigenous collaborators, including capacity and capability and other positive outcomes for Indigenous communities engaged in the research. Semi-quantitative indicator-based approaches have been used in the adaptive governance literature to determine the presence or absence of intended governance attributes [28]. Evaluation of the CAGF further contextualises findings from this systemic research design by identifying patterns and causal explanations of differences between cases or case studies [10].

## 3. Results

The CAGF is undergoing initial development, implementation, and evaluation in an NHMRC-funded national Indigenous multi-centre trial—the Flash Glucose Monitoring Study (FlashGM Study).

The FlashGM Study received NHMRC funding and ethics approval in early 2020. Governance structures, diagrams and arrangements began formulation in 2020, with early data collection in the form of indicator-based approaches for the CAGF commencing in early 2021. Other systematic evaluative processes are still in development including interview questions for the contextual approach and variables relating to location over time and across different sites for the dynamic approach. It is anticipated that all evaluative measures will be in operation by the end of 2021. A further manuscript outlining the early implementation of the CAGF in the FlashGM Study will be forthcoming.

## 4. Discussion

In 2011, Meriam Professor Kerry Arabena and David Moodie through The Lowitja Institute highlighted the lack of corresponding health improvements and outcomes for Indigenous Australians despite decades of research and medical interventions. They reflected that traditionally “the researcher identifies the priorities—a “top-down” approach. All too often this has meant that the interests of researchers do not coincide with the needs of Aboriginal and Torres Strait Islander communities” [43] (p. 533). In 2015 Gunngari/Kunja Professor Roxanne Bainbridge and colleagues stressed there was still a need to work collaboratively and listen more closely to the voices of Indigenous people, highlighting the flow of benefits as a strategic imperative that once again must include “identification of research priorities and planning, monitoring and evaluation components” if research is to truly benefit Indigenous people in Australia [7] (p. 2). Clearly, even in the era of ethical guidelines, improvements in health outcomes and benefits of research for Indigenous Australians do not reflect an increased attention placed on ethical research.

Indeed, despite successive iterations of ethical guidelines, the transformation of Indigenous health research has not transpired, and fears remain that the guidelines did not, and still do not, go far enough towards empowering Indigenous self-determination [1]. Within the documents, tensions remain relating to Western scientific traditions regulating Indigenous values and standards, which continue to pose dilemmas for individual research projects. In the face of ongoing ambiguity and in lieu of broader societal change in Australia, we posit that the missing link in the current environment for individual research projects, especially in mainstream institutions, is a framework of empowerment for Indigenous voices and priorities, centred around elevating principles of Indigenous governance.

The CAGF that has been outlined in this paper provides a replicable, holistic model of research governance that prioritises equity in the Indigenous health research agenda. As we have highlighted, traditional research governance arrangements hold inherent tensions resultant from competing governance influences, whereas Indigenous governance successfully melds together different forms of governance, i.e., people, power and relationships, with more traditional management and governance technicalities. In our view, this means that it is an ideal form of governance for the context of Indigenous health research. While highlighting the importance of Indigenous governance for improved outcomes that reflect Indigenous priorities may not be breaking any new ground, we posit that this Framework, adopting normative and practical elements from the adaptive governance literature as a vehicle for allowing this to occur, is indeed new. This governance Framework, through continual cycles of analysis and feedback adjusts research arrangements towards a new desired system logic; it guards against default to traditional, less helpful, governance arrangements for Indigenous health research.

## 5. Conclusions

If history is a guide, without new approaches that are attentive to the structural constraints entrenched within the dominant Western paradigm of ‘doing research’, we will still be calling for more Indigenous control, leadership, and perspectives in health research in another 10 years. We will be discussing the latest iteration of ethical guidelines and lamenting the lack of Indigenous representation in ethics review processes. We might also have finally found the tipping point of existing Indigenous leaders’ capacity to be all things ‘Indigenous’ for every research project and every organisational folly into Indigenous spaces in a system that fails to understand the inherent value these leaders provide.

While the creation of ethical guidelines has generally been viewed as a significant first step towards broader transformational change and improved research practices, almost 35 years since they were promulgated, ethical guidelines have yet to establish any effective mechanisms for ensuring that research adheres to diverse community expectations within the parameters of mainstream research conventions. A framework for research project governance, then, appears to be an overlooked, yet integral element for realising the ideals and recommendations of the 1986 conference and indeed, reflects the positioning of the AIATSIS Code of Ethics (2020). Recent social movements that have highlighted the structural nature of power imbalances and privilege in mainstream organisations present new opportunities for changes to governance regimes and for empowering different methodological approaches. In response to these power imbalances and privilege, it is necessary that we centre and empower Indigenous governance that is national in scope, but regional in its community focus for ensuring ethical conduct and practice in Indigenous health research.

If we are to fully realise what Ian Anderson first argued in the 1990s—that it is the researched, not the researchers, who are the primary benefactors of any inquiry, in a way that minimises the risk of research for the communities involved—a new framework that attempts to address some of the repeated failures of the last 35 years is long overdue [44]. We present the CAGF as a practical and analytical framework that recognises a history of struggle, empowers Indigenous governance and voice and develops the idea of ethics for Indigenous health research.

## Figures and Tables

**Figure 1 ijerph-18-07943-f001:**
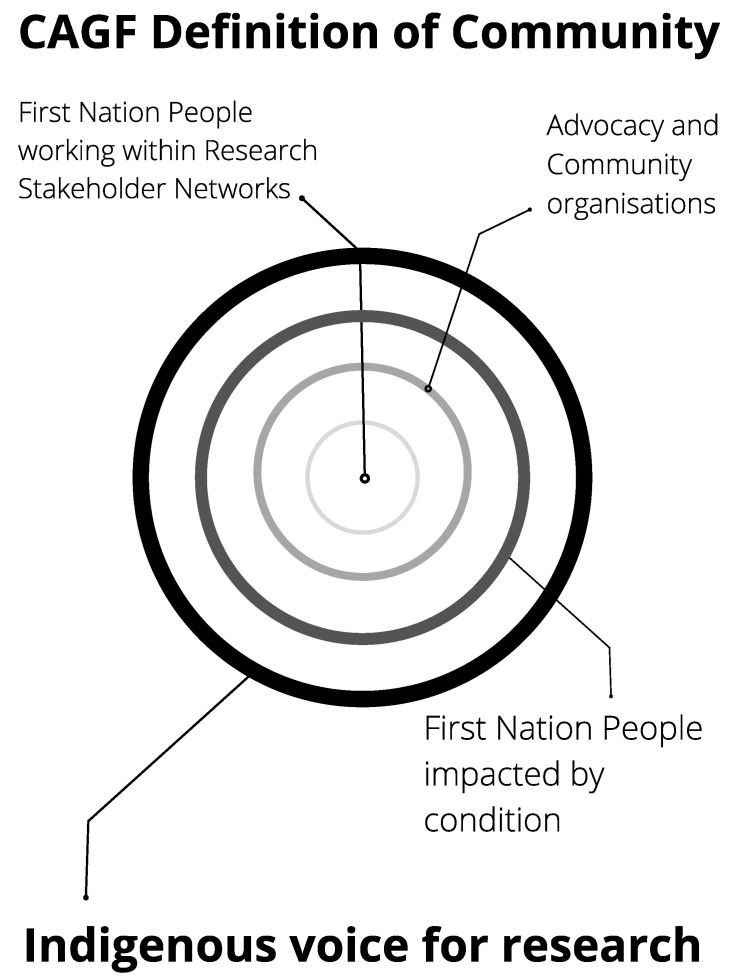
Contextual definition of community: purposeful processes of engagement, feedback and learning within a complex, dynamic and everchanging ecosystem.

**Table 1 ijerph-18-07943-t001:** Definition of key terms for the Culturally Adaptive Governance Framework.

Key Terms	Definition
Governance	Governance is the operational and control mechanisms of a defined system that holds to account people and decisions made relating to ethics, risk, consent and administration that in turn define the overall governance system.
Governance System	A set of individual and organisational actors that play a central role in decision-making and policy processes.
Governance Modes	The various forms through which governance can be realised.
Polycentric Governance	A self-organising governance system that has multiple centres of decision making and is coordinated by an overarching system of rules, rather than being imposed by one powerful actor as might be the case in a strictly hierarchical system where coordination is imposed from the top.
System Logic	The beliefs and values, socially and historically constructed, composed of symbols and material practices, by which individuals and organisations give meaning to their activities.
Indigenous Governance	A reflection of Indigenous culture, values and traditions expressed as a desire for contemporary self-determination and a need to interface this with broader governance structures of the (Australian) state.
Adaptive Governance	An analytical and practical framework for the holistic management of complex problems and governance environments.
Indigenousself-determination	Indigenous people’s right to freely pursue social, economic and cultural development as outlined in Article 3 of the United Nations Declarations on the Rights of Indigenous People.
Critical Allyship	A practice of learning and unlearning that guides people in positions of privilege to evaluate and re-evaluate their work with marginalised people through understanding health inequalities are produced within in system of structural constraints.
Equity *	Social justice or fairness; equity is an ethical concept that is grounded in principles of distributive justice.
Ethics	The values and moral principles that form the basis for decision making and conduct in relation to the impact or consequences those moral actions have on all stakeholders.

* A full discussion on distinctions between ethical theories is beyond the scope of this paper; however, we believe that current ethical guidelines were formulated on a very narrow understanding of ethics and view alternative models relating to teleological, consequentialist ethics to be more appropriate in this context.

**Table 2 ijerph-18-07943-t002:** Governance styles and their sub-functions—adapted from the governance sub-functions and governance properties in the three governance styles table in Pahl-Wostl (2015) [10].

Governance Sub-Functions	Hierarchical Style	Market Style	Network Style
Policy Framing	Prescribed by regulation; Expert judgement of problem identification; Focus on prescriptions and command and control instruments	Problem identification based on profitability, cost consideration, market failure; Focus on pricing and market-based instruments	Broad process on problem identification encompassing different perspectives; Focus on voluntary agreements
Knowledge Generation	Technocratic focus; Only technical experts involved	Knowledge serves to increase competitive advantage	Knowledge generation as part of group building process; Different types of knowledge acknowledged; Broad sharing of knowledge
Resource Mobilisation	Engage actors with political power; Tax; governmental budgets for financing	Engage actors with market power; Investment	Mobilise broad stakeholder support; Voluntary financing
Conflict Resolution	Jurisdiction; Legal procedures	Survival of the fittest; Compensation payments	Mediation; Aim for consensus
Rule Making	Political parliamentary process; Jurisdiction and formal procedures for rule extension if needed	Negotiations on prices; As few rules as possible	Broad negotiation of and deliberations on rules; Malleable rules open to renegotiation
Monitoring and Evaluation	Compliance with regulation and quantifiable standards; Rigid in terms of learning	Cost–benefit calculations; Rapid changes in individual strategies if needed to increase profitability	Participatory; Reflection on agreed goals; Openness to adaptive approaches—change negotiated
Leadership	Prescribed by formal rules; Command and control	Determined; Delegating and enabling	Often emergent in a process; Coaching and supporting
Representativeness	Elected representatives; Technical experts on problem domain	Access for all market players	All voices heard, openness of process; those affected participate in decision-making

**Table 3 ijerph-18-07943-t003:** Systemic Research Design—adapted from the typology of methods for comparative case study research in water governance, based on Pahl-Wostl and Lebel (2009) [42].

**Systemic Research Design**	**Structure**	Social networks	Diagrams of actor relationships (e.g., power, influence, authority, communication)
		Organisational	Diagrams of responsibilities and accountability relationships compared
		Systems	Diagrams of governance and other system components
	**Dynamics**	Transitions	Set of variables about same location at different times (e.g., reform process)
		Pathways	Pathways of change in different locations
	**Context**	Questions	Responses to a common set of analytical questions
		Narrative	Integrated descriptions of a governance regime
	**Indicator Based**	Checklist	Presence/absence of governance attributes
		Scoring	Ordinal scale measure of governance attributes

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
