# Peer review of "Culturally Adaptive Governance—Building a New Framework for Equity in Aboriginal and Torres Strait Islander Health Research: Theoretical Basis, Ethics, Attributes and Evaluation"

_ijerph, 2021, doi:10.3390/ijerph18157943_

Round 1

Reviewer 1 Report

The manuscript provides an articulation of the theoretical frameworks underpinning the development of a culturally adaptive governance model which is relevant for researchers working across fields of discipline and study designs in the field of Aboriginal and Torres Strait Islander health. The paper offers an example of the culturally adaptive governance model being implemented.

The paper highlights the timeline and considerations incorporated in the development of Aboriginal and Torres Strait Islander health research ethics and the lack of articulation in how to apply these ethics guidelines. The culturally adaptive governance model aims to fill this current gap and offer a process for appropriate and meaningful engagement and over sight of Aboriginal and Torres Strait Islander people in health research.

While there are no major weaknesses to the manuscript and the manuscript is very well written and drawing on relevant Indigenous literature, as an Aboriginal reviewer I have made two minor suggestions for the authors to incorporate to further articulate considerations for this work.

I suggest expanding the "Critical Allyship" articulation to engage with Indigenous literature acknowledging some of the problematic applications of the concept. An example could be work such as that by Sullivan-Clarke, A (2020) Empowering Relations: An Indigenous Understanding of Allyship in North America. Or Bennett, B (2019) Acknowledgements in Aboriginal Social Work research; How to counteract neo-colonial academic complacency in bok Disrupting Whiteness in Social Work.

I also suggest incorporating the Aboriginal Health and Medical Research Council’s (NSW) ethical guide into the “Ethical Framework for the CAGF” section.

Reviewer 2 Report

This paper proposes a new framework for conducting research relating to the health of Aboriginal and Torres Strait Islander peoples. The Culturally Adaptive Governance Framework aims to overcome existing power imbalances and improve equity by changing the nature of research governance. This is important work that has the potential to change research practice at the most fundamental level.

I have several comments the authors may wish to consider.

1) Given the nature of the work, I think it is important to understand who the authors are, their subject position with respect to this work. For example, which authors are Indigenous? What are their disciplinary backgrounds? How did they come to be involved in this work? Etc. This information would be useful to include.

2) The number of Indigenous researchers has been growing, and there are now more and more Indigenous CIAs/project leaders who control research funding.  It is not clear to me where they sit within this framework.  It would be worth exploring this issue, as I think it raises the potential for ambiguity and for multiple – and possibly conflicting – roles.  I don’t think the presence of Aboriginal and/or Torres Strait Islander project leaders eliminates or diminishes the relevance of the CAGF, but I think it is important to indicate how it applies in this case. In addition, some readers may not consider how the concept of critical allyship applies to Indigenous researchers, so it might be worth exploring this explicitly.  As another example, a given Aboriginal or Torres Strait Islander researcher leading a project may or may not belong to the community of relevance, but I’m not sure this is addressed.

3) The word ‘community’ as it is used in the manuscript might be taken to suggest something that is geographically defined. It would be useful to be explicit about the existence of non-geographically defined communities. A related point is that, for some communities (whether or not they are geographically defined), it can be difficult to identify who ‘speaks’ for the community. The process of determining ‘who speaks’ is obviously beyond the scope of the paper, but it might be worth mentioning it as a critically important – and sometimes contested – issue.

4) The paper talks about key aspects of the CAGF, but I’m not sure after reading it what exactly it looks like, in concrete terms. How is it implemented? What do you actually do?  Etc. Perhaps this is still to be developed through the FlashGM Study, but I was expected something more explicit and tangible. In any case, while it seems obvious (to me, at least) that appropriate governance should be among the very first steps in the research process, it is equally clear that this is impeded by a lack of funding to support early engagement. The authors may wish to comment on this as a potentially major barrier to implementation.

5) There is no discussion about where this works sits internationally, with respect to other Indigenous populations or other populations that are subject to power differentials. What, if any, similar approaches have there been? For example, could OCAP in Canada provide a useful model for comparison?

6) Initiatives to improve Indigenous health research processes often have implications for other population groups and, indeed, for the population as a whole. I know that is not a rationale for the CAGF, but the authors might want to explore the possibility of this framework having broader applicability.

7) There are several sentence fragments, especially in the first half of the manuscript. It would be worth getting someone who has not been involved in the drafting to cast an eye over it.

Reviewer 3 Report

Congratulations to the research team on this important work on culturally adapted research governance. The paper provides an excellent framework and approach for researchers undertaking work with and for Aboriginal and Torres Strait Islander peoples. This is a significant contribution and one likely to be highly valued by researchers seeking to design and deliver high impact, culturally safe research which benefits Aboriginal and Torres Strait Islanders. I appreciated the opportunity to review this work and look forward to seeing it in print.    

Author Response

Thank you very much for you review and support of our paper and framework.

Round 2

Reviewer 2 Report

I am satisfied with the changes made.